# Degradation of Mechanical Properties of A-PET Films after UV Aging

**DOI:** 10.3390/polym15204166

**Published:** 2023-10-20

**Authors:** Marius Vasylius, Artūras Tadžijevas, Deivydas Šapalas, Valentinas Kartašovas, Jolanta Janutėnienė, Pranas Mažeika

**Affiliations:** 1Marine Research Institute, Universiteto Av. 17, 92294 Klaipeda, Lithuania; arturas.tadzijevas@ku.lt (A.T.); deivydas.sapalas@ku.lt (D.Š.); 2Department of Engineering, Klaipeda University, Bijūnų st. 17, 91225 Klaipeda, Lithuania; pranas.mazeika@ku.lt

**Keywords:** A-PET films, UV irradiation, tensile stress, degradation of properties, impact energy

## Abstract

In 2018, the European Commission adopted the European Strategy for Plastics in a Circular Economy, which outlines key actions to reduce the negative impact of plastic pollution. The strategy aims to expand plastic recycling capacity and increase the proportion of recycled materials in plastic products and packaging. Using recycled plastic can save 50–60% energy compared to virgin plastic. Recycled PET can be used in the production of A-PET films, which are predominantly used in thermo-vacuum forming for food packaging. Storage conditions can influence the mechanical properties of polymer materials. This work presents changes in the mechanical properties of A-PET films after UV irradiation. An experimental investigation of the UV aging of A-PET films was conducted in a UV aging chamber. The specimens were exposed to a UV radiation dose rate of 2.45 W/m^2^ for 1, 2, 4, 8, 16, 24, 32, and 40 h. UV measurements were also taken on a sunny day to compare the acceleration of UV irradiation in the UV aging chamber. Mechanical tensile tests were performed on two different three-layer A-PET films (100% virgin and 50% recycled). The tensile strength and relative elongation of the A-PET films were determined, and the work required to break the film was calculated. The total consumed work was divided into the work needed for elastic and plastic deformations. A study of the UV aging of A-PET films confirmed a significant effect on the films, including a loss of plasticity even after brief exposure to solar irradiance. The results of the puncture impact test further confirmed the deterioration of the mechanical properties of A-PET material due to exposure to UV radiation, with a greater effect observed for the recycled material.

## 1. Introduction

The European Strategy for Plastics in a Circular Economy [1] notes that plastics and plastic-containing products are designed for greater durability, reuse, and high-quality recycling. Polyethylene terephthalate (PET) is a widely used thermoplastic polymer that belongs to the polyester family of polymers. It is used to make water bottles, transparent films, textile fibers (fabrics), and more. PET was first synthesized in the early 1940s, and its popularity grew rapidly with the development of synthesis methods. Among other plastics, polyethylene terephthalate is notable for being durable and relatively strong, able to be made clear, colorless, or colored (high-density polyethylene is difficult to make completely transparent), almost impermeable to gas, inexpensive to produce, and fully recyclable because its polymer chains do not break down during recycling.

The application of PET in the bottle bottling industry started in the 1970s with the advancement of moldable resins. The PET material used to create the bottles is a semi-crystalline copolymer with a glass transition temperature of approximately 76 °C and a melting temperature of approximately 250 °C. Its density ranges between 1.3 and 1.4 g/cm^3^, and it has a minimum internal viscosity of 0.7 dL/g.

PET films are widely used as a thermoformable food packaging material. Therefore, the prevalence of this substance is extremely wide. A change in its properties is a subject of study for many researchers. Some of them study the degradability of this polymer using microbiological methods to develop effective methods for the destruction of this substance, in some cases for the primary decomposition of this substance with the additional use of UV radiation; meanwhile, other scientists study the depreciation of the properties of primary or recycled PET material, which defines the storage conditions and the possibilities of high-quality use of the material [2,3,4]. These studies are particularly relevant for the purpose of using all the collected recycled PET for further use. A comprehensive examination of the mechanical, thermal, rheological, morphological, and other properties of composite materials made from recycled PET is presented in reference [5]. Amorphous polyethylene terephthalate (A-PET) is a thermoplastic film produced through extrusion from polyethylene terephthalate. A-PET is known for its excellent transparency, gloss, mechanical properties, and thermoforming capabilities. It provides an effective barrier against oxygen, nitrogen, and carbon dioxide, making it ideal for modified atmosphere packaging (MAP). A-PET can be used in contact with food and is approved by various food safety organizations in the European Union. Its antibacterial properties also make it a common choice for pharmaceutical packaging. Products and packaging made from A-PET can be recycled and reused to create the same films.

The degradation phenomena that occur during the life cycle and processing of PET are discussed in reference [6]. The solar radiation that reaches the Earth’s surface has wavelengths ranging from 295 nm to 2500 nm. It is classified into two categories: UV-B (280–315 nm), with an energy of 426–380 kJ/mol; and UV-A (315–400 nm), with an energy of 389–300 kJ/mol. Fortunately, the most energetic part of UV-B from 280 to 295 nm is filtered by the stratosphere and does not reach the Earth’s surface. UV-A is less harmful to organic matter than UV-B. The visible spectrum consists of electromagnetic waves from 400 nm to 760 nm and infrared radiation from 760 to 2500 nm [7]. The changes in the chemical structure of the polymer were studied using Fourier transform infrared spectroscopy and Raman spectroscopy in reference [8].

Many reactions can occur during exposure to sunlight, including chain decomposition due to thermal degradation between the vinyl and carboxyl chain ends, photodegradation of methylene groups leading to irreversible effects, degradation of polymer chains, and changes in the clarity of PET (yellowing) due to substances used in the synthesis and production process. Exposure to light and air causes PET photooxidation (photochemical aging). It is known that PET absorbs UV waves with wavelengths ranging from 300 nm to 330 nm. This leads to the degradation of a thin surface layer, which limits oxygen diffusion and superficial light absorption [9]. In reference [10], the optical energy gap of un-irradiated and irradiated PET samples at different wavelengths was determined using UV–Visible spectra. The molecular bond structure was analyzed before and after UV-irradiation. Thus, changes in the molecular weight distribution and morphology of PET can result from temperature and photooxidation phenomena, affecting the mechanical and thermal properties of the material.

The impact of various forms of radiation on the properties of PET is analyzed in reference [11]. It has been found that the structure of PET can be altered through chain-degradation reactions and the formation of new chemical bonds, leading to changes in its properties. As a result, many researchers have focused on the effect of physical aging on the mechanical properties of PET, using various methods to characterize both morphological changes and physical properties [12]. In addition to equipment for measuring mechanical properties (tensile strength, impact strength, and fatigue analysis), calorimetric analysis, FT-IR (Fourier transform infrared) molecular spectrometry, X-rays, and nuclear magnetic resonance (NMR) are also used.

The changes in the glass transition temperature (Tg) of PET during aging were analyzed in reference [13]. Over a period of 670 days, the glass transition temperature of aged bottles in a dark room increased from 74 °C to 78 °C, and in a sunny outdoor location, it increased from 74 °C to 87 °C. It was also observed that the endothermic glass transition became sharper as PET aged. Indoor in the dark room, the melting temperature of aged bottles decreased from 249.8 °C to 243.1 °C, and in a sunny outdoor location, it decreased from 249.8 °C to 245.6 °C.

The physical aging of semi-crystalline PET was studied using differential scanning calorimetry (DSC) at temperatures below the glass transition temperature (25 °C and 45 °C) in reference [14]. The results showed that after 120 days at 25 °C, the glass transition temperature increased slightly from 73.26 °C to 73.63 °C, and at 45 °C, it increased significantly from 73.26 °C to 86.33 °C.

In reference [15], the anisotropy of the stress–optical properties of PET films under uniaxial stress was investigated. In paper [16], the tensile properties of three-layer A-PET films with different middle layer compositions were studied. It was found that increasing the amount of recycled plastic from 20% to 100% reduced the tensile strength by 12–16%. In reference [17], the plasticity and impact properties of thermoplastic materials were studied through numerical analysis using the finite elements method, and a comparison was made with experimental results.

The plasticity of materials can be determined through impact tests. The puncture impact test of low-density polyethylene (LDPE) and linear low-density polyethylene (LLDPE) was performed, and the force versus deformation was investigated in reference [18]. The authors found that the average peak force values of LLDPE film were about 14% greater than those of LDPE film.

In reference [19], three different types of recycled high-density polyethylene were analyzed and compounded with virgin medium density polyethylene (MDPE) in an extruder. The impact resistance results showed that the 50/50 ratio produced results like those of MDPE. In reference [20], six different methods for determining the ductile/brittle transition temperature were analyzed, one of which was the Elastic energy ratio. In this method, the peak load energy (E_m_) is compared to elastic energy (E_el_.).

In practice, recycled A-PET is used to produce food packaging films. From the production of the film to its use for packaging, it must be stored under appropriate conditions. Practical observations showed that the films can change their mechanical properties, becoming more brittle, making the molding process more challenging.

The objective of this study is to determine the mechanical properties of A-PET films after aging in a UV aging chamber for various lengths of time, comparing virgin A-PET and recycled R_A-PET materials.

## 2. Materials and Methods

Materials. Experimental tests of three-layer virgin and recycled A-PET films were conducted. Two types of materials were tested: 100% virgin A-PET film (clear color) and recycled R_A-PET in (black color), where the inner layer of the film consists of 50% recycled and 50% virgin material, and the outer layer is made entirely of virgin A-PET. The thickness of the R_A-PET (black) film is 0.5 mm, and the thickness of the virgin A-PET (clear) film is 0.35 mm.

Tensile test. Mechanical tensile testing of thin plastic films was performed in accordance with the standard ASTM D882-2:2004 [21]. Specimens of 250 mm in length and 25 mm in width were cut from the A-PET roll. Both non-aged and UV-aged films were tested using a Zwick/Roell Z020 universal testing machine (Figure 1a). Each test was repeated three times, and the mean values of the results were calculated. The following parameters were measured on the testing machine: tensile strength, relative elongation at maximum force and film breakage, modulus of elasticity, and the work required to break the film. From the measured total energy, the work consumed for elastic and plastic deformation was determined.

The plastic film specimens were aged in an Atlas UV test UV aging chamber (Figure 1b) following ASTM G154 procedures [22]. A 40W UVA-351 fluorescent lamp was used, which simulates the UV energy in sunlight passing through a window glass. Both types of films were aged in the chamber for 1, 2, 4, 8, 16, 24, 32, and 40 h. The UV irradiance dose rate received by the films from the fluorescent lamps was 2.45 W/m^2^. The machine has 24 specimen holders, with a total area of 6848 cm^2^. The distance between machine UV lamps and specimens is 52 mm. The machine is also capable to perform condensation weathering tests.

The dose rate is the amount of UV energy falling on an object per unit area. The temperature in the UV aging chamber was maintained at 60 °C. The total UV doses received by the different films are shown in Table 1.

Puncture impact test. The puncture impact testing of A-PET films was performed using a drop weight testing machine (Figure 1c), measuring the force dependency over time when the puncture probe hit the film specimen. The puncture impact test was performed according to ISO 7765-2 standard [23]. Test specimens were cut into 80 × 80 mm size squares. During the test, specimens were securely clamped with camping rings which have an inside diameter of 40 mm. The test was performed with hemispherical, polished, and hardened striker with a diameter of 20 mm. The main parameters of performed tests were as follows: drop weight, 9.421 kg; drop height, 987 mm; penetrator weight, 0.19 kg; impact velocity, 4.4 m/s; potential energy, 96.51 J.

In accordance with the recommendations of the ASTM D882-2 standard [21], and based on the relative elongation of the film measured during the preliminary break test (up to 20%), the crosshead speed of the testing machine was set to 12.5 mm/min, and the distance between the grippers was 125 mm. The test was stopped when the tensile load decreased by more than 80%. The ambient conditions during the tests were as follows: 21.1 °C ambient temperature; 43% relative humidity.

UV irradiation. The amount of UV energy in sunlight under real-world conditions was estimated and compared. UV measurements were taken on June 17 2020 on a sunny day with no clouds. The measurements were taken from 10:30 to 17:00 in three directions—southeast, south, and southwest—with the instrument tilted 45° towards the sky from the horizon. The geographical coordinates of the UV measurement location were 55.7274° latitude and 21.1260° longitude. The UV dose rate was measured using a UV test calibration radiometer UVA at 351 nm, and a one-day graph is shown in Figure 2.

## 3. Results and Discussion

As shown in Figure 2, the results of the UV dose rate measurements indicate that the maximum amount of UV energy from sunlight in the normal direction occurs for 2 h per day.

The maximum measured UV dose rate was 2.91–2.93 W/m^2^. The total daily UV irradiance received from sunlight was 12.57 Wh/m^2^ in the southeast, 13.27 Wh/m^2^ in the south, and 13.68 Wh/m^2^ in the southwest. The one-hour average was 0.52 Wh/m^2^ in the southeast, 0.55 Wh/m^2^ in the south, and 0.57 W/m^2^ in the southwest. The recalculated UV irradiance is presented in Table 1, allowing us to see the amount of UV the film would have received if it had been naturally exposed to sunlight.

A-PET specimen strips were inserted into the grippers of the testing machine using rubber gaskets to prevent significant deformation of the specimen strips and slippage of the specimens within the gripper. The fixed specimens in the grippers of the Zwick/Roell Z020 testing machine are shown in Figure 3.

The specimen in the grippers during the puncture impact test is shown in Figure 4.

The A-PET film specimens after the tensile and puncture impact tests are shown in Figure 5. As the data indicate, the specimens made from virgin material (clear film) began to break brittlely after being exposed to UV radiation for 4 h, while the specimens composed of recycled material (black film) exhibited brittle fracture even after just 1 h of UV aging.

The curves of tensile stresses and strain obtained during the tests are shown in Figure 6 and Figure 7. From these curves, we can see that even with the slight aging of the A-PET films in the UV aging machine, the material loses its plasticity. The R_A-PET material loses its plasticity faster than virgin material. As we can see in Figure 7, when exposed to UV, polymer bonds decompose faster and lose the important plasticity property of the film.

The dependencies of tensile strength and strain at maximum force on UV irradiance are shown in Figure 8 and Figure 9, respectively. These dependencies show that the tensile strength of R_A-PET films is 20% higher than that of virgin films. After more than 8 h of UV exposure (or 19.6 Wh/m^2^), the tensile strength does not change; it remains almost the same; i.e., the R_A-PET films are about 59.2 MPa and the virgin A-PET films are about 49.4 Mpa. The relative elongation (strain) of maximum tensile strength films also has a limiting UV aging time. Figure 9 shows that after UV aging for 1–4 h (up to 9.8 Wh/m^2^), the relative elongation of films increases, but when they are UV-aged for more time, their relative elongation begins to decrease, the material becomes harder, and its plasticity decreases. In graphs, error bars indicate the standard deviation of the measurements and show how the set of experimental results tend to be close to the mean.

The breaking point of the specimens also clearly demonstrates the impact of UV aging on A-PET material. The tensile stress of the breaking point of the recycled materials increased from 15.5 Mpa to an average value of 54.43 Mpa after just 1 h of UV exposure (4.9 Wh/m^2^) and remained similar even after prolonged aging (Figure 10).

The values of the relative elongation at the breaking force show a strong influence of UV on the plastic properties (this dependence is shown in Figure 11). We observed a strong decrease of the R_A-PET (black) films’ relative elongation from 16,84% to an average value of 5.61% after 1 h of UV exposure (2.45 Wh/m^2^), and one from 20.64% to an average value of 5.19% for A-PET (clear) film after 2 h of UV exposure (4 Wh/m^2^).

The values of the relative elongation at the breaking force demonstrate the significant impact of UV on the plastic properties, as shown in Figure 11. There is a sharp decrease in the relative elongation of the R_A-PET (black) films—that is, from 16.84% to an average of 5.61% after just 1 h of UV exposure (2.45 Wh/m^2^). Similarly, the relative elongation of the A-PET (clear) film decreases from 20.64% to an average of 5.19% after 2 h of UV exposure (4 Wh/m^2^).

The effect of UV aging on plastics is clearly demonstrated by the dependence of the work required to break the film, as shown in Figure 12. In this graph, we can see that even after exposing the film to UV for 1 h (2.45 Wh/m^2^), the work required to break the film decreases by about 2.26 times for recycled material and about 2.77 times for virgin material.

By dividing the work required to break the specimens of film into elastic and plastic zones, we can analyze the details more closely. As we can see in Figure 13, even after aging the films, a little more work is required to perform elastic deformations, as the film has hardened after receiving UV radiation. After 4 h (19.6 Wh/m^2^), these values are almost unchanged, with the work required to perform elastic deformations for the virgin film being about 1.38 J, and that for recycled material being 2.43 J. However, the work required to perform plastic deformations decreases significantly, as we can see in Figure 14. The values of plastic deformations for unaged recycled film decrease from 4.42 J to 0.33 J and remain similar for aged film. The virgin film decreases from 3.29 J to 0.23 J, a drop of 13.4 times for recycled films and 14.3 times for virgin films.

Puncture impact tests were performed to determine the nature of fracture during impact, which can be either ductile or brittle. The Zwick/Roell drop weight tester recorded the force during the short impact process, as shown in Figure 15. The force–time graphs reveal the first part of the impact, i.e., when the puncture probe hits the specimen. The magnitudes of force and deflection were calculated to determine the energy required to penetrate A-PET film specimens.

The force and deflection dependence curves of 100% virgin A-PET are presented in Figure 16 and Figure 17. The force and deflection dependence curves of R_A-PET (50% recycled + 50% virgin) films at different accelerated UV aging times are presented in Figure 18. These graphs demonstrate that when the films are exposed to UV aging, the maximum force required to penetrate the specimens of virgin A-PET films decreases by 15% and R_A-PET by 25%. This indicates a significant degradation of the film’s strength after UV aging. The degradation of the film’s plasticity is also evident, as the deflection of the film at the maximum force decreases to 25% of virgin A-PET films and even to 60% of R_A-PET films.

The moment of the initial crack appearance can be observed in tests of unaged A-PET films, where the force suddenly decreases slightly when the crack opens. However, this moment could only be clearly identified in unaged films. With even a little UV aging, the films become more brittle, and the testing machine is no longer able to capture the moment of the initial crack opening. Thus, only the graphs of maximum force and energy versus UV irradiance, presented in Figure 19, Figure 20 and Figure 21, will be considered in the further analysis of the test results.

Puncture resistance is a measure of the maximum force or energy required to penetrate an A-PET film. Almost twice as much force is needed to penetrate R_A-PET films compared to 100% virgin films (about 1150 N and 630 N for unaged films, respectively). In Figure 19, we can see maximum force decreases by about 15–25% when the UV irradiance reaches 98 Wh/m^2^ (or after 40 h of accelerated UV aging).

The dependencies of the energy at the maximum force and the total energy required to penetrate the specimens on UV irradiation were also obtained, and they are presented in Figure 20 and Figure 21. After the films receive 20 Wh/m^2^ of UV irradiation (or 8 h of accelerated UV aging), the total energy required to penetrate the R_A-PET films decreases by 60%, and that of the virgin films by 50%. After longer UV irradiation, the energy required to penetrate the specimens decreases a little more (by about 5%) and remains similar. The total energy of both film types aged with 98 Wh/m^2^ UV irradiation was almost equal (3.8 J and 3.0 J), although the energies of the new films differed by almost twice the amount (13.1 J and 6.6 J).

This experimental research clearly shows that the film rapidly and significantly loses its plasticity when it is even slightly exposed to UV light. As a result, the use of film in thermal molding can become challenging. Products such as various packaging materials or other products may crack or tear during the manufacturing process, due to the loss of the film’s plasticity after exposure to UV light.

## 4. Conclusions

Two types of A-PET films were analyzed: A-PET (pure virgin material) and R_A-PET (50% recycled and 50% virgin material). The films were aged in a UV chamber for 1 to 40 h; in other words, up to 98 Wh/m^2^ dose, with 2.45 W/m^2^ irradiation intensity. The results confirmed the significant impact of UV on the mechanical properties of the films, particularly in the case of the R_A-PET material. The degradation of the properties was best demonstrated by the calculated work required for elastic and plastic deformations. It was found that exposing the A-PET film to UV light for just 1–2 h (up to 5 Wh/m^2^ under 2.45 W/m^2^ irradiation intensity) caused a significant loss of its plasticity properties. Thus, films used in the thermoforming of packaging should be protected from direct sunlight.

According to an analysis of chemical transformations in the PET under UV irradiation, there are few main causes of the changes in the mechanical properties. The main cause seems to be a crystalline amount increment in a PET. Authors who have analyzed this process in depth using three different methods—DSC thermogram, FTIR-ATM spectrum, and density changes—have demonstrated that there is direct relationship between crystalline amount and UV irradiation dose [13]. Moreover, this explains a plasticity decrement during UV irradiation and a tensile strength increment. Another researcher group states that a decrement of mechanical properties in PET can be induced via carboxyl formation under UV irradiation in environments that partially have oxygen [24].

Puncture impact test results showed that the examined PET films are extremely sensitive to photodegradation under UV exposure. By contrast, virgin PET is bit less sensitive; it shows 50% of the fracture energy decrement after the first 8 h of artificial UV exposure in a UV aging chamber, and its approximately 20 Wh/m^2^ semi recycled PET film, marked R_A-PET, shows higher sensitivity to UV irradiation (after the same exposure, fracture energy decreased by 60%).

A-PET films used in the thermoforming of packages must be protected from direct sunlight when they are stored. Since the glass transition temperature increases after UV aging [13,14], the technological process of thermoforming food packages and bottles changes, and low-quality packages are obtained.

Since recycled PET is also used in personal protection products (e.g., face shields, glasses, etc.), long-term use of these products outdoors in the sun (i.e., under UV exposure) makes them not only less transparent but also more fragile, so they may break after mechanical impact and injure the face.

## Figures and Tables

**Figure 1 polymers-15-04166-f001:**
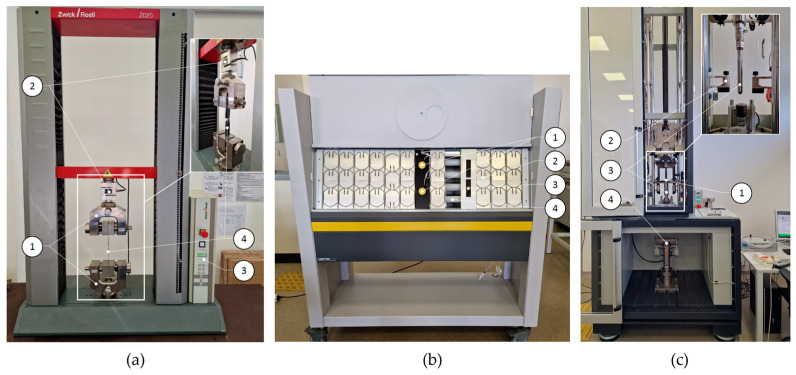
Testing equipment. (**a**) Universal testing machine Zwick/Roell Z020: 1—specimen-holding clamps “GRIPS”; 2—20 kN force cell; 3—machine controller; 4—specimen. (**b**) UV-aging chamber Atlas UVTest: 1—UV sensors; 2—specimen holder; 3—specimen; 4—UV lamp. (**c**) Drop-weight-testing machine Zwick/Roell Amsler HIT200F: 1—variable weight carriage; 2—safety pins; 3—puncture probe; 4—pneumatic specimen holding grips.

**Figure 2 polymers-15-04166-f002:**
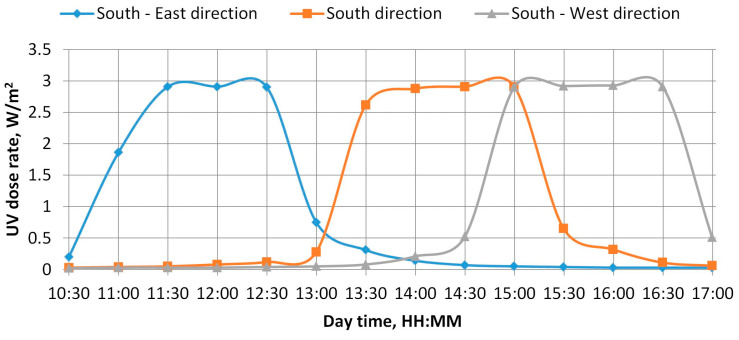
Results of UV dose rate measurements.

**Figure 3 polymers-15-04166-f003:**
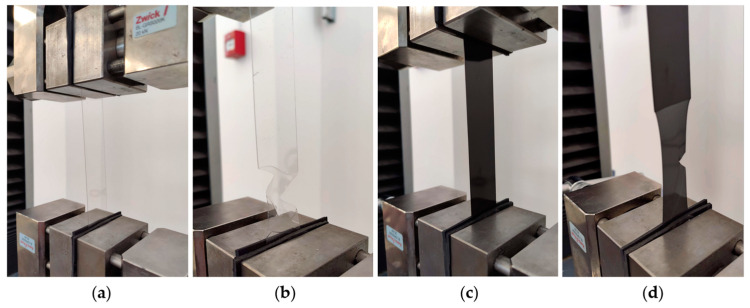
Tensile tests of non-aged virgin A-PET and recycled R_A-PET specimens: (**a**) non-aged virgin A-PET specimen before test; (**b**) non-aged virgin A-PET specimen after test; (**c**) recycled R_A-PET specimen before test; (**d**) recycled R_A-PET specimen after test.

**Figure 4 polymers-15-04166-f004:**
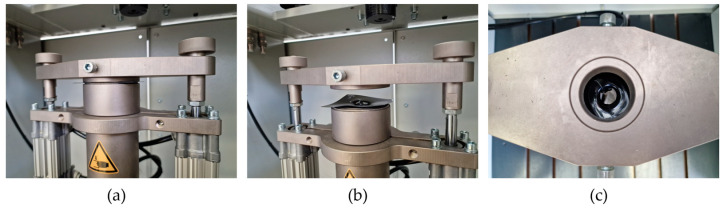
Puncture impact test of A-PET film: (**a**) before test; (**b**) after test (side view); (**c**) after test (top view).

**Figure 5 polymers-15-04166-f005:**
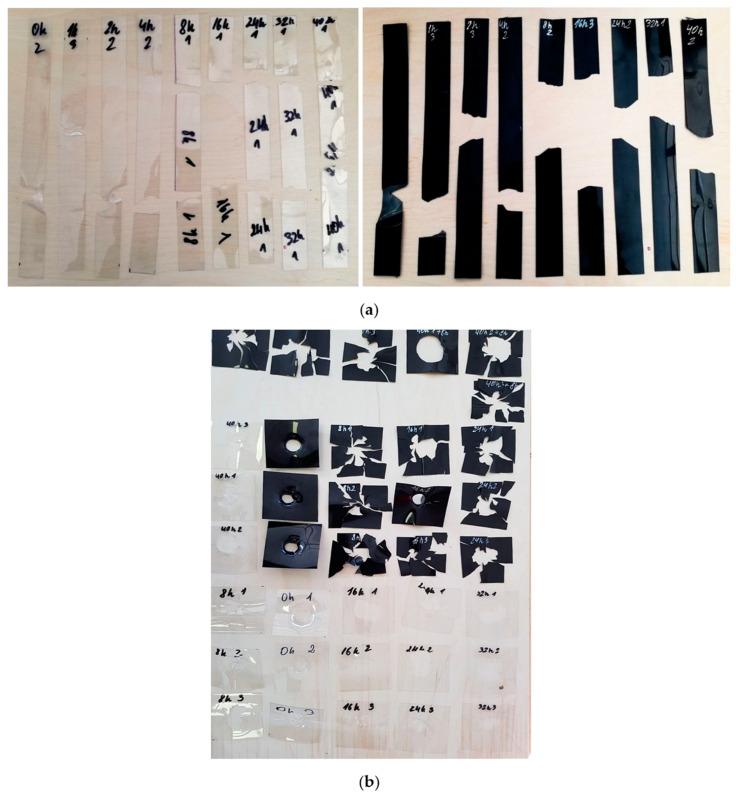
View of virgin and recycled A-PET film specimens: (**a**) after tensile test; (**b**) after puncture impact test.

**Figure 6 polymers-15-04166-f006:**
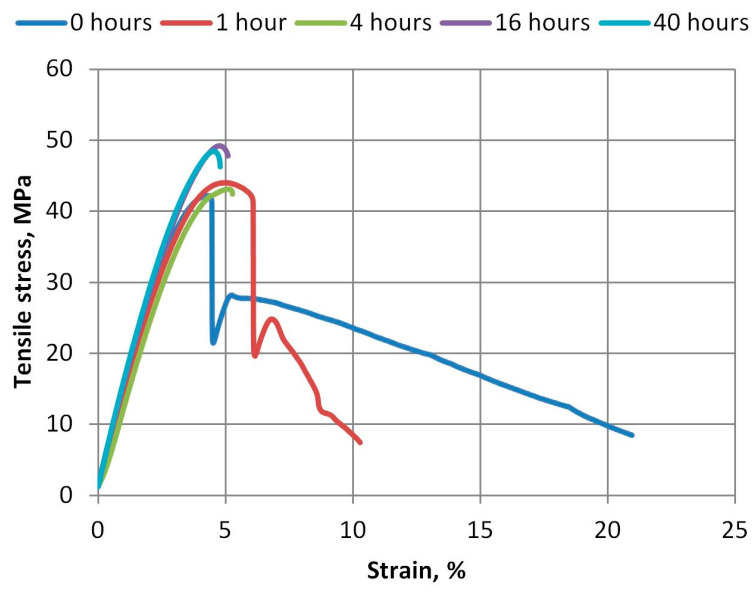
Tensile stress and strain curves of virgin A-PET films aged at different durations.

**Figure 7 polymers-15-04166-f007:**
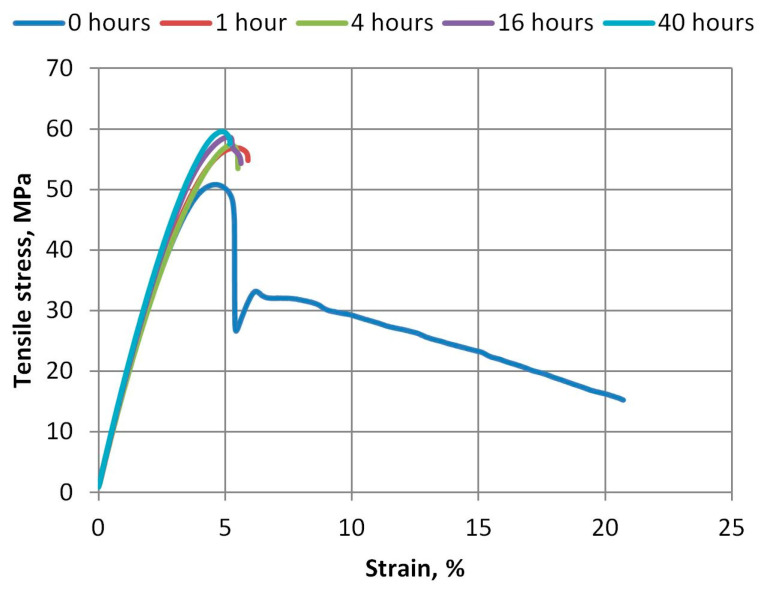
Tensile stress and strain curves of R_A-PET films aged at different durations.

**Figure 8 polymers-15-04166-f008:**
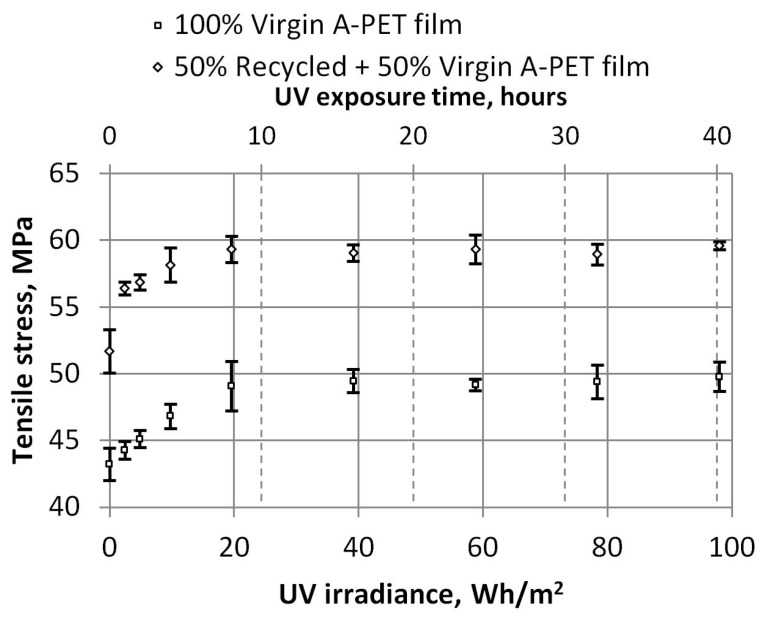
Dependence of tensile strength and UV irradiance.

**Figure 9 polymers-15-04166-f009:**
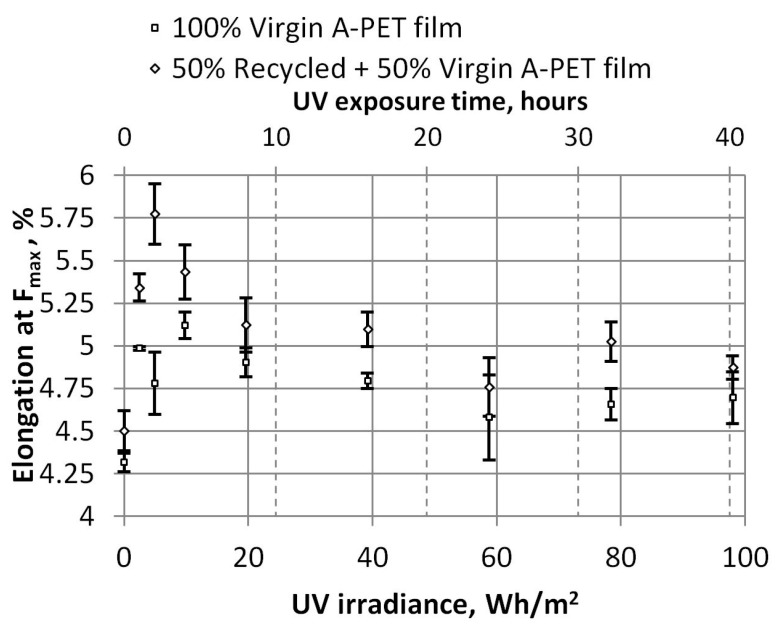
Dependence of relative elongation at maximum force and UV irradiance.

**Figure 10 polymers-15-04166-f010:**
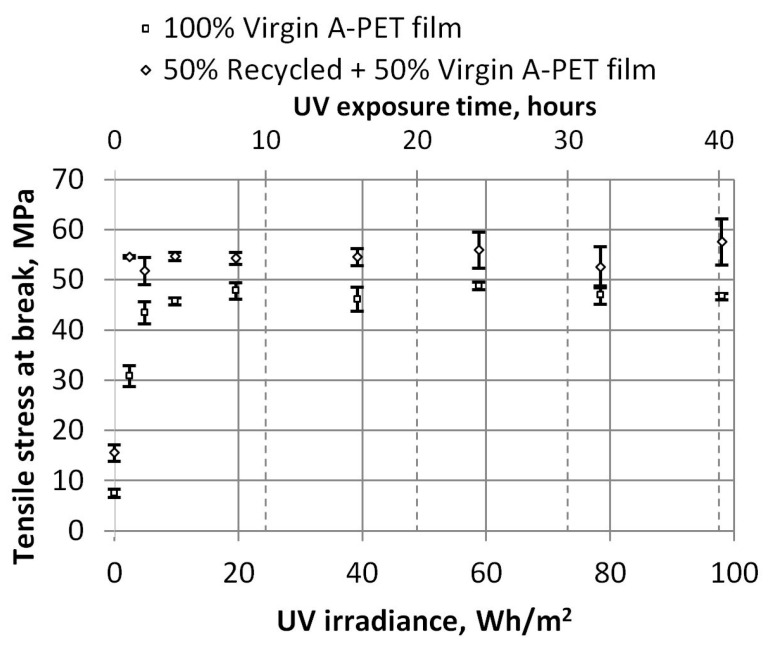
Dependence of tensile stresses at break and UV irradiance.

**Figure 11 polymers-15-04166-f011:**
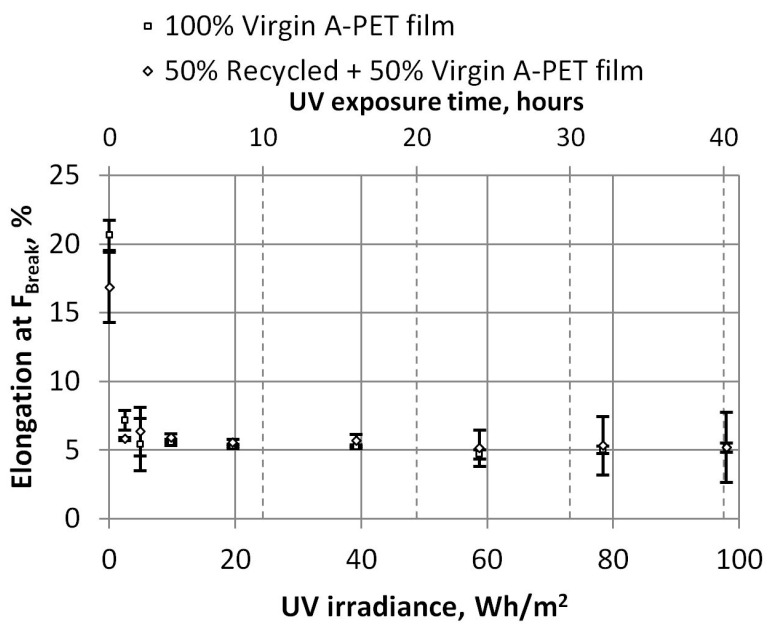
Dependence of relative elongation at break and UV irradiance.

**Figure 12 polymers-15-04166-f012:**
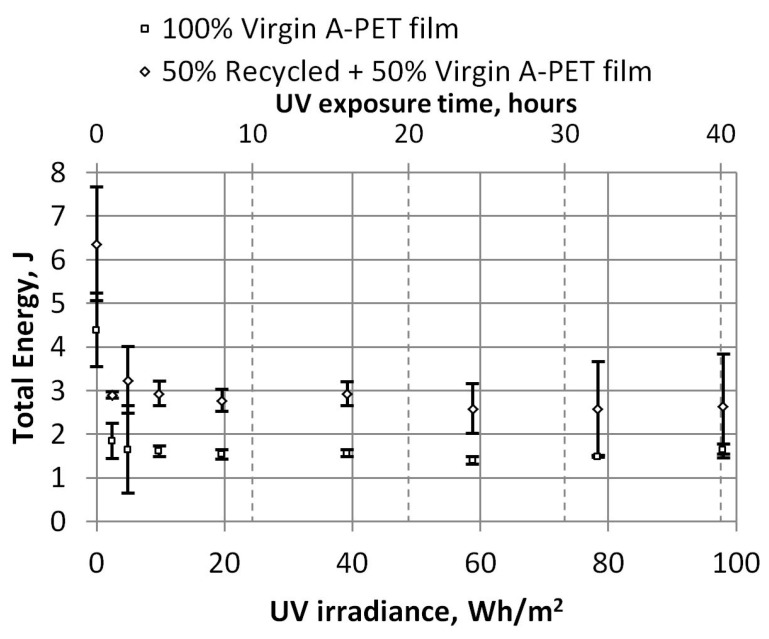
Dependence of UV irradiation on the total work required to break the film (tensile test).

**Figure 13 polymers-15-04166-f013:**
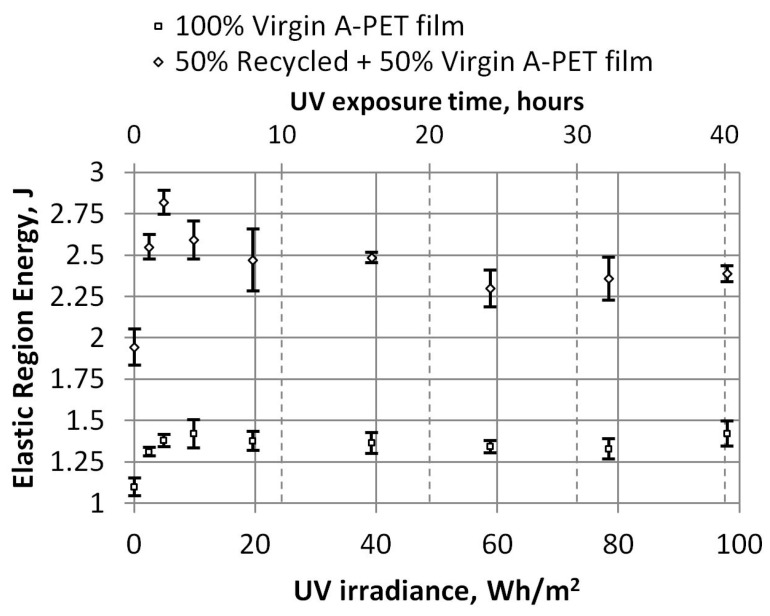
Dependence of UV irradiation on elastic region energy (tensile test).

**Figure 14 polymers-15-04166-f014:**
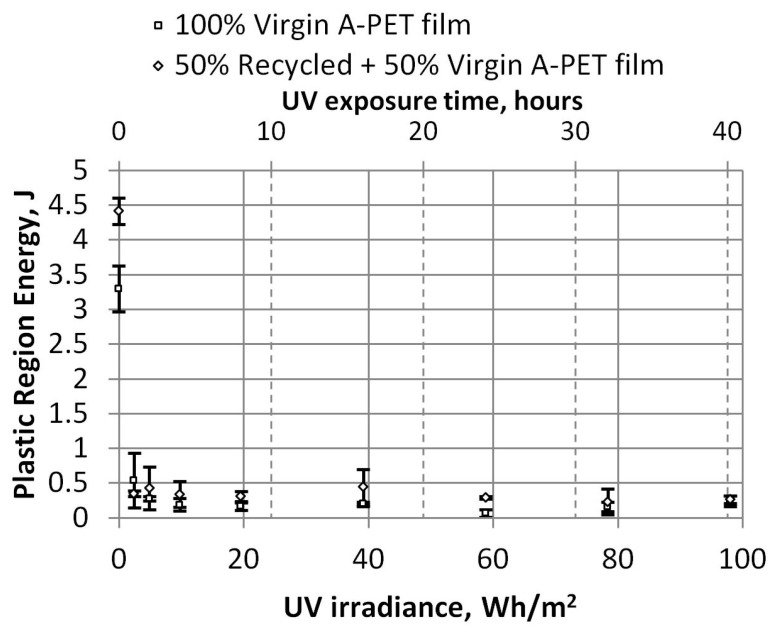
Dependence of UV irradiation on plastic region energy (tensile test).

**Figure 15 polymers-15-04166-f015:**
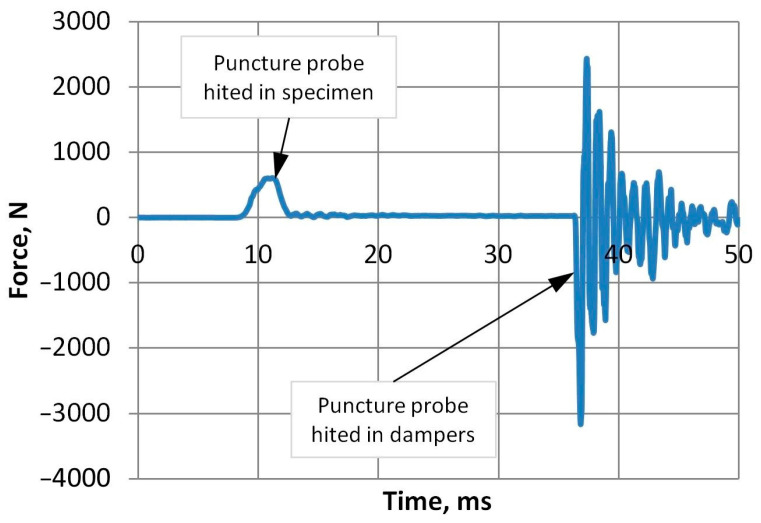
Force versus time diagram (100% virgin material, not UV-aged).

**Figure 16 polymers-15-04166-f016:**
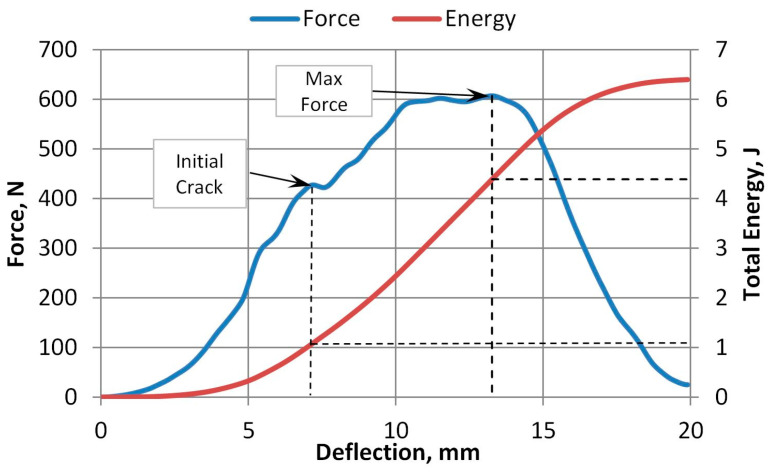
Force and total energy versus deflection plots (specimen is 100% virgin film not UV-aged).

**Figure 17 polymers-15-04166-f017:**
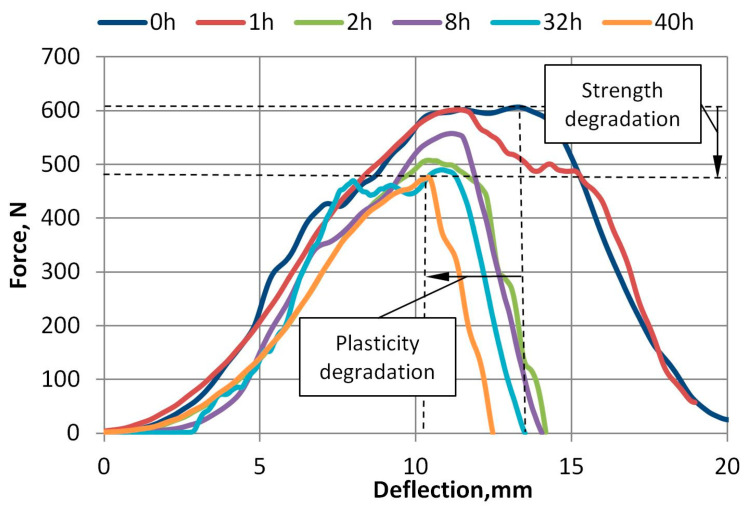
Force versus deflection plots after UV aging of 100% virgin A-PET films.

**Figure 18 polymers-15-04166-f018:**
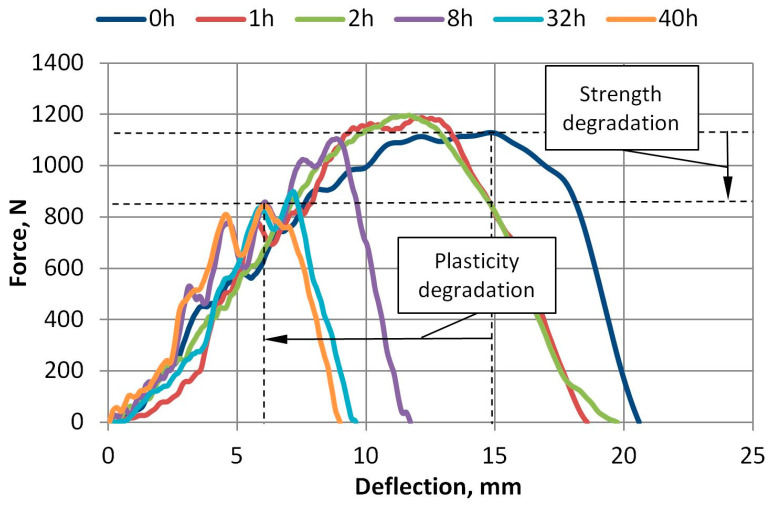
Force versus deflection plots after UV aging of R_A-PET films.

**Figure 19 polymers-15-04166-f019:**
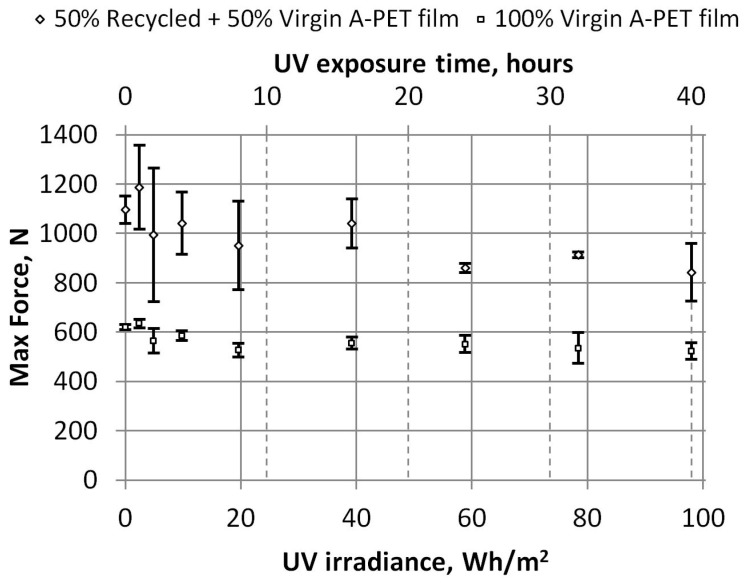
Dependence of maximum force on UV irradiance.

**Figure 20 polymers-15-04166-f020:**
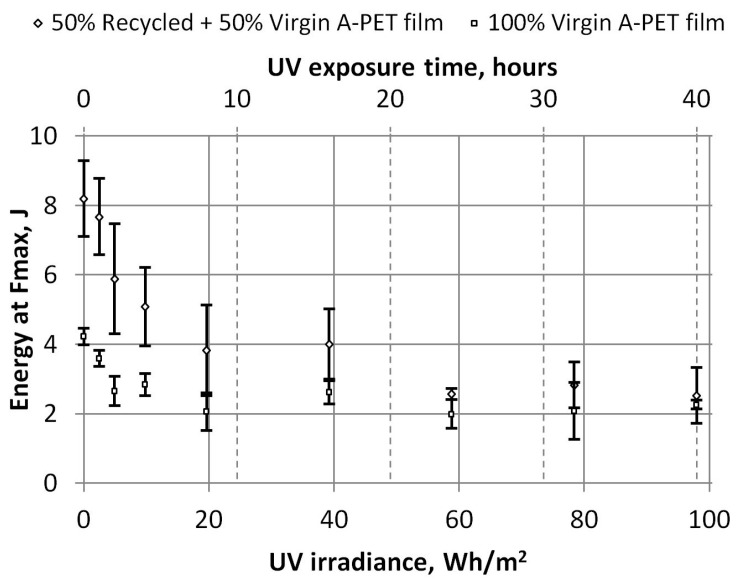
Dependence of UV irradiation on energy at maximum force consumed to break A-PET films.

**Figure 21 polymers-15-04166-f021:**
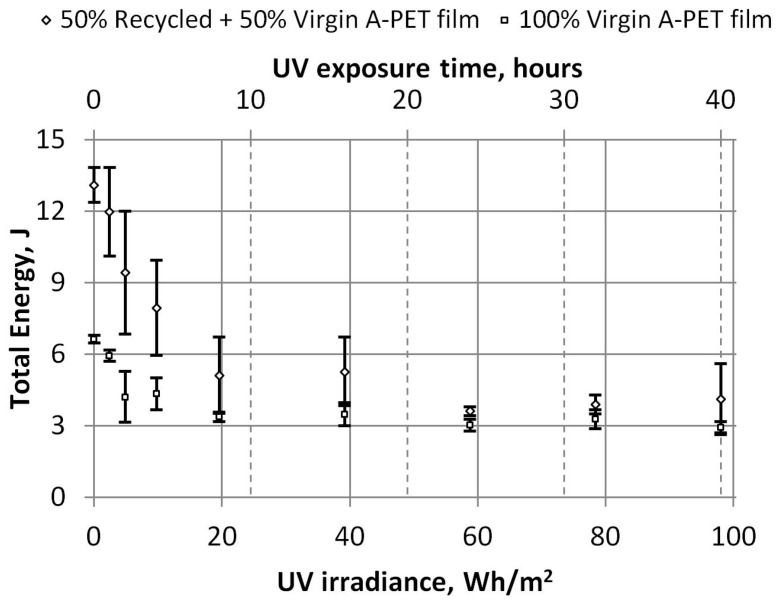
Dependence of UV irradiation on the total energy consumed to break A-PET films.

**Table 1 polymers-15-04166-t001:** Accelerated UV aging hours and total UV irradiance.

Film Type	Accelerated UV Aging, Hours	Accelerated UV Irradiance, Wh/m^2^	Natural Solar UV Aging, Hours
1	1	2.45	4.3
2	2	4.9	8.6
3	4	9.8	17.2
4	8	19.6	34.4
5	16	39.2	68.8
6	24	58.8	103.2
7	32	78.4	137.5
8	40	98	171.9

## Data Availability

The data presented in this study are available on request from the corresponding author.

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
