# Peer review of "Degradation of Mechanical Properties of A-PET Films after UV Aging"

_polymers, 2023, doi:10.3390/polym15204166_

Round 1

Reviewer 1 Report

The article "Degradation of Mechanical Properties of A-PET Films after UV Aging" provides valuable insights into the impact of UV aging on the mechanical properties of A-PET films, particularly in the context of recycling and sustainability. However, there are some drawbacks and areas for improvement:

1.The article briefly mentions the UV aging chamber and exposure conditions but lacks a detailed description of the methodology. Readers would benefit from more information on the setup, UV radiation levels, and the testing procedures followed.

2. While the article mentions exposure times ranging from 1 to 40 hours, the data presentation is relatively limited. Providing graphical representations of the changes in mechanical properties over time would enhance the clarity and impact of the findings.

3. Understanding the microstructural changes in the A-PET films due to UV aging is essential. The article could benefit from microscopic or spectroscopic analyses to correlate mechanical property changes with structural alterations.

4. While the article mentions the impact of UV aging on both virgin and recycled A-PET films, a more in-depth comparative analysis between the two types of films would be beneficial. This could include a detailed discussion of why recycled A-PET appears to be more affected.

5. The article highlights the negative effects of UV aging on A-PET films, but it could delve further into the practical implications. For instance, how might these findings influence the choice of materials for food packaging in real-world applications? Are there potential strategies to mitigate UV-induced degradation?

6. Although the article mentions the European Strategy for Plastics in a Circular Economy, it could benefit from a more extensive review of relevant literature on UV aging of polymer materials. This would provide context and enable readers to understand how this study contributes to the existing body of knowledge.

7. The article lacks a concise summary or concluding remarks section that synthesizes the key findings and their significance. Such a section can help readers grasp the main takeaways from the study.

Some sentences are quite long and complex, which may make the text less accessible to a general audience. Simplifying the language and sentence structure would improve overall clarity.

Reviewer 2 Report

The manuscript by Vasylius et al. describes the effect of UV irradiation on the mechanical properties of virgin and recycled amorphous-PET films. The PET films were irradiated in a UV chamber for various time periods and characterized through mechanical properties analysis, including tensile tests and impact tests, to evaluate the effect of UV radiation. Overall, the work conducted does not meet the criteria for publication in Polymers, and I would not recommend its publication in its present form. The following concerns should be addressed:

1.     The introduction section is poorly described; there are many references on PET degradation under the influence of irradiation that should be included in the manuscript.

2.     Photos of the testing equipment are not included in the manuscript, but they should be.

3.     The data from the tensile and impact tests are limited. A systematic study with a larger number of samples is needed.

4.     Standard deviation and error bars should be provided.

5.     An explanation is needed for why the elongation at break was very low for the PET film (15% to 20% for non-UV-irradiated films).

6.     While the decrease in elongation at break supports the degradation of PET, the reason for the increase in tensile strength at break should be described in the manuscript.

7.     The obtained results should be compared with literature values of mechanical properties, and the phenomena occurring during UV irradiation should be described.

8.     The conclusion section appears weak. How did the authors differentiate the behavior of virgin and recycled PET towards UV light? Recycled PET already has lower mechanical properties than virgin PET. A fair comparison should be provided.

The manuscript by Vasylius et al. describes the effect of UV irradiation on the mechanical properties of virgin and recycled amorphous-PET films. The PET films were irradiated in a UV chamber for various time periods and characterized through mechanical properties analysis, including tensile tests and impact tests, to evaluate the effect of UV radiation. Overall, the work conducted does not meet the criteria for publication in Polymers, and I would not recommend its publication in its present form. The following concerns should be addressed:

1.     The introduction section is poorly described; there are many references on PET degradation under the influence of irradiation that should be included in the manuscript.

2.     Photos of the testing equipment are not included in the manuscript, but they should be.

3.     The data from the tensile and impact tests are limited. A systematic study with a larger number of samples is needed.

4.     Standard deviation and error bars should be provided.

5.     An explanation is needed for why the elongation at break was very low for the PET film (15% to 20% for non-UV-irradiated films).

6.     While the decrease in elongation at break supports the degradation of PET, the reason for the increase in tensile strength at break should be described in the manuscript.

7.     The obtained results should be compared with literature values of mechanical properties, and the phenomena occurring during UV irradiation should be described.

8.     The conclusion section appears weak. How did the authors differentiate the behavior of virgin and recycled PET towards UV light? Recycled PET already has lower mechanical properties than virgin PET. A fair comparison should be provided.

Round 2

Reviewer 2 Report

The manuscript has undergone satisfactory revisions and is now ready for acceptance for publication in Polymers

The manuscript has undergone satisfactory revisions and is now ready for acceptance for publication in Polymers